# Pre-emptive Action Revision by Environmental Feedback for Embodied Instruction Following Agents

**Jinyeon Kim**[1,2,*]    **Cheolhong Min**[1,*]    **Byeonghwi Kim**[1]    **Jonghyun Choi**[1,†]

[1]Seoul National University          [2]Yonsei University

jinyeonkim@yonsei.ac.kr, {cheolhong.min,byeonghwikim,jonghyunchoi}@snu.ac.kr

https://pred-agent.github.io

**Goal:** *Cook two slices of potato and place them on a clean plate.*

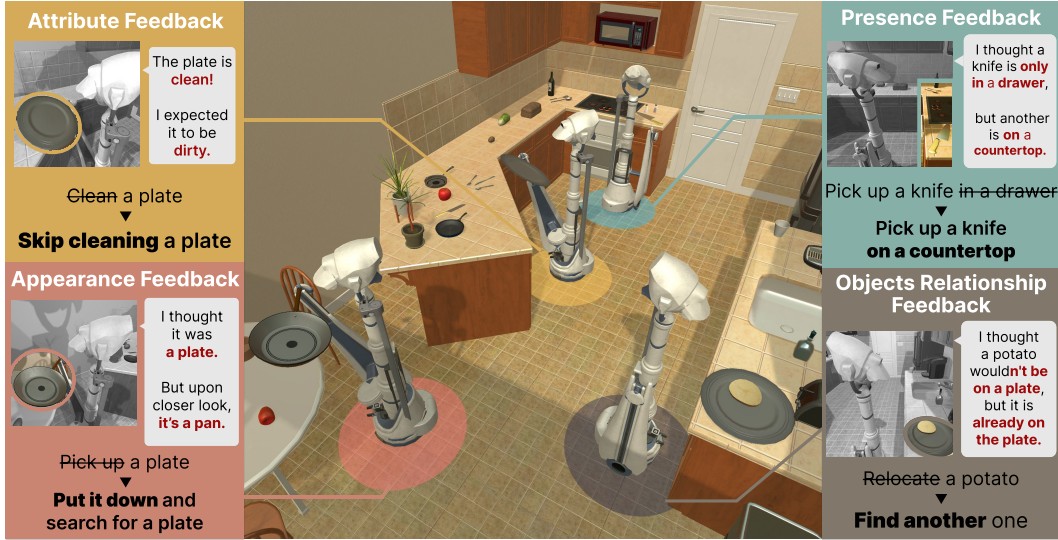

Figure 1: **Overview of the proposed method (PRED).** The agent needs to adapt to unexpected environmental state changes that are different from the environment that it initially conceived: it skips cleaning an already clean plate, puts down a pan when a plate is needed, grabs a knife from the countertop without further searching, and ignores moving a potato that is already on its destination plate. By the proposed PRED powered by LLMs, the agent effectively achieves the goal of serving two slices of potato on a clean plate without making mistakes.

**Abstract:** When we, humans, perform a task, we consider changes in environments such as objects' arrangement due to interactions with objects and other reasons; *e.g.*, when we find a mug to clean, if it is already clean, we skip cleaning it. But even the state-of-the-art embodied agents often ignore changed environments when performing a task, leading to failure to complete the task, executing unnecessary actions, or fixing the mistake after it was made. Here, we propose Pre-emptive Action Revision by Environmental feeDback (PRED) that allows an embodied agent to revise their action in response to the perceived environmental status *before it makes mistakes*. We empirically validate PRED and observe that it outperforms the prior art on two challenging benchmarks in the virtual environment, TEACh and ALFRED, by noticeable margins in most metrics, including unseen success rates, with shorter execution time, implying an efficiently behaved agent. Furthermore, we demonstrate the effectiveness of the proposed method with real robot experiments.

**Keywords:** Replanning, Environmental Feedback, Brain plasticity, Embodied AI

---

*Equal contribution. †Corresponding author.

8th Conference on Robot Learning (CoRL 2024), Munich, Germany.

# 1 Introduction

Building robotic assistants that understand natural language and surroundings and perform tasks has long been an ambitious research goal. Recent advances in computer vision [1, 2, 3] and natural language processing [4, 5, 6, 7] have significantly contributed to the development of such agents, enabling them to carry out various tasks [8, 9, 10, 11, 12, 13, 14] in diverse environments [15, 16, 17].

Typically, agents plan actions based on initial environmental states [18, 19], but environmental discrepancies between expected and actual states often cause task failures due to misperception or incomplete environmental exploration. Unlike artificial agents, humans and animals adapt to environmental changes through *brain plasticity* [20, 21, 22], revising behaviors based on experience to avoid mistakes. In light of this, we ask: *Can embodied agents benefit from similar adaptability?*

Drawing inspiration from neuroscience, we propose Pre-emptive Action Revision by Environmental feeDback (PRED), an instruction that follows the embodied agent that adjusts its behavior by perceiving environmental discrepancies such as environmental feedback and revising action plans using large language models (LLMs). For environmental discrepancies, considering that object perception plays an important role in numerous embodied tasks, we particularly focus on four distinct environmental discrepancies related to objects: 1) object presence [23, 24], 2) object appearance [25], 3) object attributes [26, 27], and 4) object-object relationships [28, 29].

To address these discrepancies, we propose four components: 1) Dynamic Target Adaptation (DTA) that dynamically modifies navigation targets using object presence discrepancies, 2) Object Heterogeneity Verification (OHV) that verifies whether an interacted object is intended by examining object appearance discrepancies between the initial and subsequent perceptions of the object, 3) Attribute-Driven Plan Modification (APM) that modifies the original state-changing actions, such as cleaning an object, using object attribute discrepancy, and 4) Action Skipping by Relationship (ASR) that refrains agents from taking unintended actions using object relationship discrepancy. Previous approaches [26, 30] revise their original plans after taking actions and encountering failures. But failure may result in irreversible consequences *e.g.*, splitting milk on the floor. Here, we propose *preemptively action revision policy for the initially planned action sequences* to avoid nonrecoverable failures such as irreversible state transitions as in the mentioned example [12, 13, 31].

We empirically validate PRED in two challenging benchmarks, TEACh [13] and ALFRED [12], for embodied instruction following, as well as with real robot experiments. We observe that PRED outperforms prior arts notably in most metrics, including the main metric, the unseen success rate.

# 2 Related Work

**Embodied instruction following agents.** Developing agents that understand natural language to achieve goals is a daunting challenge. To develop such agents, previous benchmarks [32, 33, 34] focus on language comprehension and navigation. For example, [32] tasks a robot with inferring the next steps based on natural language, and [33] requires reaching a specified destination. However, these focus mainly on navigation without object interaction, limiting their application to more complex tasks (*e.g.*, preparing breakfast using a toaster and coffee machine).

To tackle tasks beyond navigation, recent benchmarks [13, 14, 35, 36] include object interaction, which requires agents to understand language and manipulate objects. Early methods [37, 14, 38, 39, 40] learn to map multi-modal inputs (*i.e.*, egocentric observation, and language) to the corresponding actions and object locations. However, these approaches often need large training datasets for high performance, which is expensive and sometimes unfeasible.

To address the data-scarcity issue, recent approaches [41, 42, 26, 30] use deterministic algorithms, such as $A^*$ or FMM [43], for accurate obstacle-free navigation on semantic spatial maps [44, 18, 42, 30], significantly improving their performance. Inspired by this, we also exploit [43] for navigation.

**Planning and revising using large language models.** Recent work [30, 45, 46, 47] leverages LLMs to revise their plans after agents perform the actions and fail. However, some failures are

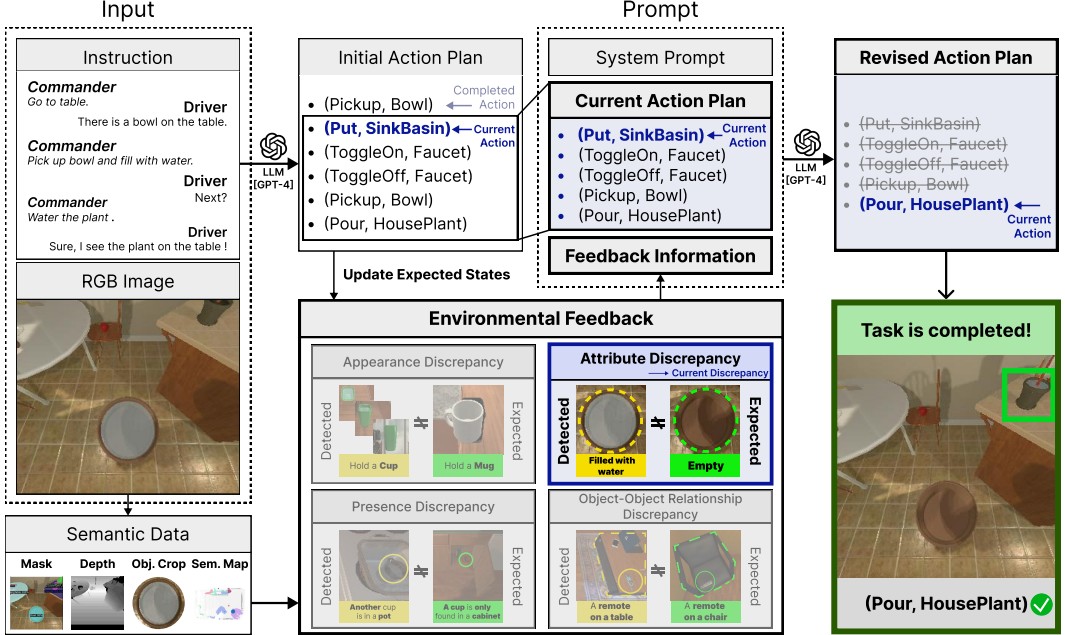

Figure 2: **Workflow of the proposed Pre-emptive Action Revision by Environmental feeDback (PRED).** We first generate an initial plan based on the language instruction using a LLM and predict initial and target object states to achieve a goal. Simultaneously, we predict the environmental states based on the predicted semantics. We then compare the anticipated and actual environmental states and maintain the discrepancies between them. If there are discrepancies, we use them as environmental feedback, denoted by 'Feedback Information,' to revise the original plan, denoted by 'Current Action Plan.'

irrecoverable (*e.g.*, a broken plate), causing a task failure. Although these methods revise actions after the failure, we *revise actions beforehand* to avoid unnecessary steps.

Another line of work [48, 49] tries to solve task planning and failure correction by focusing on errors. However, no approach explores the possibility of skipping unnecessary actions for efficiency. Our method enables agents to adapt their actions based on the environment and improve efficiency like humans without needing detailed failure information.

To correct the plan by receiving environmental feedback, there is some work that takes advantage of ground truth feedback [50, 51, 52, 53, 30]. On the other hand, [30, 46] updates their plans, but does not focus on object-centric representation that can lead to interaction with unintended objects. [47, 54] try to address environmental discrepancies, but can only address a limited discrepancy. For the sake of space, we elaborate the details of the mentioned works in the Appendix A.

## 3 Approach

Recently proposed embodied agents use LLMs [30, 46, 55] for commonsensical reasoning and semantic spatial maps [42, 44, 18] for path planning. However, these agents often encounter scenarios where their anticipated environmental states differ from the ones they actually perceive. These scenarios can result in unintended or incorrect actions and consequently lead to inefficiency or even failure in completing tasks. Inspired by humans and animals that adapt to changing environments, we propose PRED that pre-emptively adjusts plans using environmental discrepancies as feedback. Figure 2 illustrates the workflow of PRED. Further details of PRED can be found in the Appendix C.

### 3.1 Revising Actions using Environmental Feedback

We aim to learn a policy that generates a modified action sequence for embodied agents [12, 13] by perceiving environmental changes. Specifically, we learn a function $f_s : \mathcal{V} \times \mathcal{A} \rightarrow \mathcal{A}$, that maps a visual observation set, $\{v_m\}_{m=1}^{M}$, and an ($N$-length) action sequence, $\{a_n\}_{n=1}^{N}$, into a modified ($K$-length) action sequence, $\{a'_k\}_{k=1}^{K}$. $\mathcal{V}$ and $\mathcal{A}$ denotes all possible observation and action sets.

We consider an embodied agent, a function, $\pi_\theta : \mathcal{V} \times \mathcal{X} \rightarrow \mathcal{A}$, that maps a natural language instruction, $x \in \mathcal{X}$, and the current visual observation, $v_t \in \mathcal{V}$, into an action, $a_t \in \mathcal{A}$. We expect the generated action sequence, $\{a_t\}_{t=1}^T$, to transform the initial environment state, $s_i \in \mathcal{S}$, to the desired state, $s_f \in \mathcal{S}$, where each $\mathcal{X}$ and $\mathcal{S}$ denotes a set of all possible natural language instructions and environment states and $T$ a time budget. We detail our agent's architecture in the Appendix B.

In particular, when an agent faces unexpected scenarios caused by 'differences' between inferred and observed states, referred to as 'environmental discrepancies,' PRED revises a original plan, $\{a_n\}_{n=1}^N$, by querying a large language model, $\mathcal{L}$, with a prompt, $\mathcal{P}$. $\mathcal{P}$ concatenates a system prompt, $\mathcal{P}_s$, for a general description and guide of the task, the original plan, and a feedback prompt, $\mathcal{P}_f$, which describes the discrepancy encountered as environmental feedback generated by the LLM. Then $\mathcal{L}$ receives $\mathcal{P}$ and produces a revised plan, $\{a_k'\}_{k=1}^K$ as:

$$\{a_k'\}_{k=1}^K = \mathcal{L}(\mathcal{P}) \quad \text{where} \quad \mathcal{P} = [\mathcal{P}_s; \{a_n\}_{n=1}^N; \mathcal{P}_f]. \tag{1}$$

Additional details of revising actions are provided in the Appendix C.1. To build a feedback prompt, we consider four types of environmental discrepancies caused by the presence, appearance, attributes, and relationships of objects based on visual information that occupies a large proportion of sensory information perceived by humans [56, 57]. We propose four modules for respective discrepancies: Dynamic Target Adaptation (DTA), Object Heterogeneity Verification (OHV), Attribute-Driven Plan Modification (APM), and Action Skipping by Relationship (ASR). We detail the proposed modules below and provide their examples of a system prompt, a current action plan, how the feedback prompt is generated, and an LLM output in listings in the Appendix G.

### 3.1.1   Presence Discrepancy → Dynamic Target Adaptation

When an agent is conducting a task, it may encounter scenarios where a target object is present in unexpected places. For example, in the task, 'boil a potato in the refrigerator using a pot,' the agent may expect the refrigerator, possibly containing the potato, as its navigational target. However, the agent may find another potato on different objects, such as a table. In this case, revising the action plan by adapting to environments where target objects can be easily found may improve the efficiency and effectiveness of navigation. Here, we denote by an object presence discrepancy the difference between an expected location and an actual one for a target object to be found.

To address this, we propose Dynamic Target Adaptation (DTA) which detects an object presence discrepancy and provides a feedback prompt describing it. For this, the agent first compares an inferred place of a target object, $o$, with one perceived and maintained in the agent's memory (see the Appendix B for more details), $Z_t$, at the current time step, $t$. If the agent has previously observed the target object in a different place from the inferred one (i.e., $o \in Z_t$) before reaching it, DTA returns the presence discrepancy as a feedback prompt, $\mathcal{P}_f$, indicating that the target object is not in the expected location (e.g., $\mathcal{P}_f =$ ''`The target is either directly visible to the agent or its information is already stored on the map.`'').

### 3.1.2   Appearance Discrepancy → Object Heterogeneity Verification

An agent often fails to interact with task-relevant objects due to misperceptions influenced by lighting, occlusions, and varying appearances from different viewpoints. This issue can be mitigated by examining objects from multiple perspectives. For example, an agent might mistake a cup for a mug due to object recognition errors from a far distance.

To address this issue, we propose Object Heterogeneity Verification (OHV), which recognizes appearance discrepancies and provides feedback accordingly. Here, we define an appearance discrepancy as the difference in an object's predicted identities (i.e., classes) from the appearances observed in various viewpoints. The proposed verification requires the agent to pick up objects and change its view. That is, when the agent interacts with an object, it verifies if the object is intended by comparing its predicted classes that can be different due to varying appearances from various viewpoints. These actions allow the agent to see the object without occlusion, thus encouraging it to properly identify the object (i.e., a mug) by the observations from various viewpoints.

Formally, let $c_i$ be a predicted object class from the appearance of a $i^{th}$ viewpoint of an object, where $i \in \{0, \cdots, I\}$ denotes a viewpoint index among predefined $I$ viewpoints. $i = 0$ is the viewpoint at the time of interaction (see the Appendix C.2 for more details). If the agent encounters a different predicted class (*i.e.*, $c_0 \neq c_i$) from any different viewpoint $i > 0$, contrary to the expected class (*i.e.*, $\forall i > 0 : c_0 = c_i$), OHV detects an appearance discrepancy and specifies this in a feedback prompt, $\mathcal{P}_f$, indicating that the object is not intended (*e.g.*, $\mathcal{P}_f =$ ``After checking, the picked up object's mask is detected, but the object is not the desired one...'').

### 3.1.3 Attribute Discrepancy → Attribute-Driven Plan Modification

Language instructions often lack detailed environmental descriptions, potentially causing the agent to do redundant actions or miss important actions due to unknown object attributes. Here, an *attribute* refers to the physical state of an object depending on its affordances[*] [13, 26]. We can mitigate redundant or missed actions from unknown attributes by checking the target object, $o$, its expected attribute, $\hat{\phi}_o$, and its actual observed attribute, $\phi_o$.

For example, in a task, "clean a mug and fill it with coffee," the agent expects that $o$ (*i.e.*, a mug) needs to be cleaned because it is dirty, so the expected attribute of the target can be represented as $\hat{\phi}_o = \{\text{Dirty}\}$. However, the agent might find a clean mug during exploration, where the detected target object's attribute is clean (*i.e.*, $\phi_o = \{\text{Clean}\}$). In this case, the agent can skip redundant actions in the original plan (*i.e.*, cleaning the mug first), allowing efficient task completion. We denote an attribute discrepancy as the difference between the expected and observed attributes of an object.

For this, we propose Attribute-Driven Plan Modification (APM) to detect attribute discrepancies and provide environmental feedback. When the agent detects $o$, it captures a cropped image exclusively of the target from the current view. Then, we utilize an attribute detector, $\mathcal{H}$, which takes the cropped image $v_o$ as input and predicts $\phi_o$ (see the Appendix C.3 for more details). If $\phi_o$ from $\mathcal{H}$ does not match $\hat{\phi}_o$ (*i.e.*, $\hat{\phi}_o \neq \phi_o$), APM describes this attribute discrepancy as a feedback prompt $\mathcal{P}_f$ (*e.g.*, ``The target object in the test image has been either cleaned, as determined by the similarity check.'').

### 3.1.4 Object-Object Relationship Discrepancy → Action Skipping by Relationship

Rearranging objects [10, 58] often poses challenges when the agent relocates multiple objects, potentially with the same look, of the same class, making it difficult for the agent to decide which object to move. To address this, the agent considers the current relationship, $r_o$, and the expected relationship, $\hat{r}_o$, between the target object, $o$, and its placement object, $o_p$. Here, the relationship represents the spatial relationship between $o$ and $o_p$. We write this as $r_o = (o, o_p)$.

For instance, if instructed to "place two pillows on the sofa," the agent assumes that $o$ (*i.e.*, a pillow) is not on the final place (*i.e.*, sofa) and needs to be moved. Thus, the expected relationship can be defined as the pillows that are not on the sofa, represented as $\hat{r}_o = \neg(pillow, sofa)$. However, during exploration, if the agent detects a pillow on the sofa, the current relationship becomes $r_o = (pillow, sofa)$, *i.e.*, $\hat{r}_o \neq r_o$, and a relationship discrepancy occurs. If the agent ignores such discrepancy and continues to interact with $o$, *i.e.*, meaninglessly moving the pillow from the sofa to the sofa, it may not be able to complete the task.

To mitigate meaningless relocation, we propose Action Skipping by Relationship (ASR), which detects the relationship discrepancy and provides a corresponding feedback prompt. To obtain $r_o$, we first predict the masks of objects from the current egocentric view, $\{m_i\}_{i=0}^N$, where $m_0$ denotes the mask of $o$. We then find the most 'adjacent' mask, $m_i$, of $o_p$, to $m_0$ and regard that $o$ is currently on $o_p$ (see the Appendix C.4 for more details). If the currently detected target object is already in the final place (*i.e.*, $\hat{r}_o \neq r_o$), ASR describes this relationship discrepancy in a feedback prompt $\mathcal{P}_f$, such as ``After checking, it is found that the second object has already been placed in the desired location as indicated by the interaction mask, ...''

---

[*]We use object attributes supported by AI2-THOR [15] on which our evaluation benchmarks [12, 13] run.

Table 1: **Comparison with the state of the arts in the TEACh benchmark.** The path-length-weighted (PLW) metrics are given in the parentheses for each value. The highest and second highest values per fold and metric are shown in **bold** and underline, respectively.

| Model | TfD | | | | EDH | | | |
|---|---|---|---|---|---|---|---|---|
| | Unseen | | Seen | | Unseen | | Seen | |
| | SR | GC | SR | GC | SR | GC | SR | GC |
| E.T [37] | 0.48 (0.12) | 0.35 (0.59) | 1.02 (0.17) | 1.42 (4.82) | 7.80 (0.90) | 9.10 (1.70) | 10.20 (1.70) | 15.70 (4.10) |
| JARVIS [41] | 1.80 (0.30) | 3.10 (1.60) | 1.70 (0.20) | 5.40 (4.50) | 15.80 (2.60) | 16.60 (8.20) | 15.10 (3.30) | 22.60 (8.70) |
| FILM [42] | 2.90 (1.00) | 6.10 (2.50) | 5.50 (2.60) | 5.80 (11.60) | 10.20 (1.00) | 18.30 (2.70) | 14.30 (2.10) | 26.40 (5.60) |
| DANLI [26] | 7.98 (3.20) | 6.79 (6.57) | 4.97 (1.86) | 10.50 (10.27) | 16.98 (7.24) | 23.44 (19.95) | 17.76 (9.28) | 24.93 (22.20) |
| HELPER [30] | 13.73 (1.61) | 14.17 (4.56) | 12.15 (1.79) | 18.62 (9.28) | 17.40 (2.91) | 25.86 (7.90) | 18.59 (4.00) | 32.09 (9.81) |
| **PRED** (Ours) | **19.77** (5.16) | **16.74** (8.31) | **20.99** (4.64) | **21.55** (11.03) | **21.69** (4.44) | **26.83** (7.46) | **21.71** (4.62) | **32.78** (10.39) |

## 4 Experiments

We briefly explain the benchmarks, baselines, and evaluation metrics used for the experiments (see Appendix D for details of benchmarks and baselines). We also validate our method in both simulation and real-world manipulation experiments. The experiment setup is detailed in Appendix E.1.

**Benchmarks.** We employ two challenging long-horizon instruction following benchmarks for embodied agents, TEACh [13] and ALFRED [12]. In TEACh, we evaluate our PRED in two sub-benchmarks, Trajectory from Dialog (TfD) and Execution from Dialog History (EDH). TfD requires the agent to solve long-horizon household tasks by understanding dialogs, while EDH requires performing a session-specific portion of the TfD tasks. ALFRED provides declarative instructions consisting of a goal statement and step-by-step instructions that describe how to complete a task.

**Baselines.** We compare PRED with recent state-of-the-art methods as baselines for both benchmarks. For the TEACh benchmark, we compare ours with FILM [42], DANLI [26], and HELPER [30]. For the ALFRED benchmark, we adopt HLSM [44], FILM [59], and CAPEAM [18] as baselines.

**Metrics.** The primary metric is the success rate (SR), the ratio of the completed tasks. The goal condition success rate (GC) denotes the ratio of the satisfied goal conditions. To measure efficiency, the path-length-weighted (PLW) score penalizes SR and GC by the length of the actions taken.

### 4.1 Comparison with State of the Arts

We compare PRED with prior state-of-the-art methods on the TEACh and ALFRED benchmarks summarized in Table 1 and Table 2, respectively. Additionally, we provide the results of a new valida-tion and test set in EDH, used exclusively by DANLI [26] and re-split from the original validation set, to ensure a fair comparison (see the result table in the Appendix E.2). PRED outperforms other baseline models in both benchmarks by noticeable margins in SR and GC, demonstrating its efficacy.

In the TEACh benchmark, in TfD and EDH setups, we observe that PRED outperforms the previous methods in unseen/seen environments for SR and GC, which implies the effectiveness of our proposed PRED. In addition, our model shows a larger performance gap between ours and prior art on TfD, a more changeable setup than on EDH. This may be because TfD requires more interactions with objects than EDH, which performs a specific portion of the TfD tasks. Thus, our proposed methods have more opportunities to be applied, leading to greater improvement in TfD compared to EDH.

We observe that DANLI [26] achieves better PLW scores in the EDH setup. We believe that its 3D map tracking each instance's location, including height, eliminates the need for vertical scanning in navigation, unlike our top-down 2D map. In addition, its recovery plans' effectiveness is based on a lot of human-defined plans for all exceptions, improving PLW scores.

Table 2 shows the prior arts and PRED's performance in the ALFRED benchmark with a few different settings. We include the 'Reproduced' section because the reproduced results of previous methods differ slightly from the originally reported ones. As shown in Table 1, we observe that our method outperforms the prior arts in all metrics, implying the effectiveness of the proposed components.

**Diverse initial states.** In the AL-FRED benchmark, all objects' states are static at the beginning of every task, indicating the initial states of objects are always fixed once a task for the agent to perform is given. For example, in the task of moving a cleaned mug, the initial states of all mugs in the environment are always set to be 'dirty.' However, this evaluation does not fully address environmental discrepancies caused by different initial states of objects (*e.g.*, cleaning a mug that is already clean), potentially resulting in unexpected scenarios.

To further address these discrepancies in the ALFRED benchmark, we intentionally modify the initial states of objects to have diverse ones and denote these modified ones as 'diverse initial states.' For example, in the task of moving a cleaned mug above, the agent may encounter already cleaned mugs and the agent can revise its plan for efficient and effective task completion. We observe that our PRED outperforms all prior approaches in this setting, implying that our PRED has better capability to adapt to environments with objects' diverse initial states.

Table 2: **Comparison with the state of the arts in the ALFRED benchmark.** The path-length-weighted (PLW) metrics are given in the parentheses for each value. The highest and second highest values per fold and metric are shown in **bold** and underline, respectively. The 'Reported' and 'Reproduced' sections show the methods' performances from the paper and our reproduction results. 'Reproduced w/ Diverse Initial States' shows performances in diverse initial states (Section 4.1).

| Model | Test Unseen | | Test Seen | |
|---|---|---|---|---|
| | **SR** | **GC** | **SR** | **GC** |
| **Reported** | | | | |
| HLSM [44] | 20.27 (5.55) | 30.31 (9.99) | 29.94 (8.74) | 41.21 (14.58) |
| FILM [59] | 24.46 (9.67) | 34.75 (13.13) | 25.77 (10.39) | 36.15 (14.17) |
| CAPEAM [18] | 43.69 (17.64) | 54.66 (22.76) | 47.36 (19.03) | 54.38 (23.78) |
| **Reproduced** | | | | |
| HLSM [44] | 21.32 (5.89) | 31.09 (10.39) | 31.90 (9.75) | 43.22 (15.30) |
| FILM [59] | 23.61 (15.10) | 36.90 (12.99) | 25.77 (10.58) | 35.43 (14.62) |
| CAPEAM [18] | 41.79 (18.07) | 53.93 (23.41) | 45.14 (18.79) | 52.82 (23.25) |
| **PRED** (Ours) | **46.96** (20.58) | **57.35** (24.72) | **51.40** (21.14) | **59.04** (25.52) |
| **Reproduced w/ Diverse Initial States** | | | | |
| HLSM [44] | 19.03 (5.60) | 28.21 (9.64) | 23.27 (7.59) | 31.56 (12.60) |
| FILM [59] | 15.17 (7.43) | 22.74 (12.14) | 13.89 (5.68) | 22.80 (10.26) |
| CAPEAM [18] | 24.00 (9.39) | 31.92 (14.85) | 25.18 (10.78) | 32.97 (15.96) |
| **PRED** (Ours) | **32.31** (12.48) | **42.62** (17.87) | **35.09** (15.02) | **43.27** (19.95) |

## 4.2 Ablation Study

To assess the impact of each proposed component—DTA, OHV, APM, and ASR —we conduct ablation studies and summarize the results in Table 3. Additional ablation results, including other splits in TEACh and ALFRED, are provided in Appendix F. In our experiments, no simultaneous environmental discrepancies were detected, but our PRED is designed to handle multiple discrepancies at once. Here, '($x$) *vs.* ($y$),' denotes a comparison between the $x$ and $y$ rows in Table 3.

**No DTA.** We observe that ablating DTA ((a) *vs.* (b)) leads to noticeable drops (up to $2.68\%$ in ALFRED test unseen) in all metrics for TfD, EDH, and ALFRED. We believe that our agent without DTA does not consider object presence discrepancies, causing repeated unnecessary object interaction. This can increase the chance of interaction failure and thus, reduce task success rates.

**No OHV.** Second, we ablate OHV ((a) *vs.* (c)) to assess the impact of considering differences in object appearances. Ablating OHV results in significant performance drops, with a $4.57\%$ SR decrease in the TfD valid unseen and a $3.01\%$ SR decrease in the ALFRED test unseen. We empirically observe that the agent suffers from misperceptions such as light reflection and occlusion, causing it to interact with irrelevant objects. Unlike the agent with OHV, the ablated one continues without verifying whether it interacted with the correct object, leading to task failures due to misperceptions.

**No APM.** When APM is absent from PRED ((a) *vs.* (d)), the agent may struggle to detect changes in the attributes of the object, leading to unnecessary actions or the omission of key actions. This results in performance drops in all metrics (*e.g.*, $7.98\%$ SR in ALFRED test unseen). The oversight in object attributes can cause the agent to unnecessarily change an object's state (*e.g.*, cleaning an already cleaned object) or attempt infeasible actions to the objects that are not in an available state for interaction. This increases failures and time steps, reducing the likelihood of task completion.

**No ASR.** We ablate ASR from our agent ((a) *vs.* (e)). Without ASR, the agent cannot comprehend the relationships between objects, which makes it attempt to move the objects again that already satisfy the goal relationship (*e.g.*, a *Plate* should be in *Sink*). This may lead to achieving insufficient goal states as the agent moved 'two' objects but actually, it did only 'one' object. Due to this, we observe that a $2.61\%$ SR drop in TfD valid unseen and a $4.06\%$ SR drop in ALFRED test unseen.

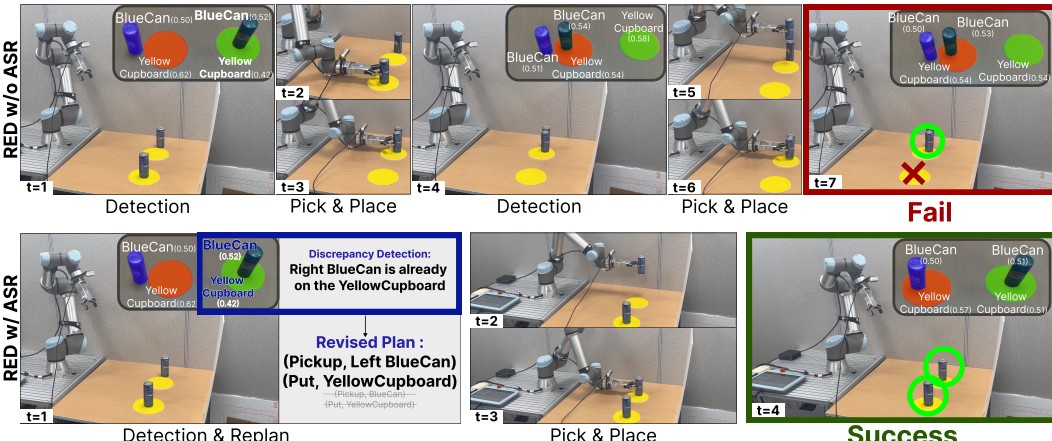

Figure 3: **Qualitative analysis of ASR in real-robot experiments.** In the leftmost figure, the agent initially checks the environment. Without the ASR method (top row), the agent follows the initial plan and repeatedly moves a blue can to the yellow cupboard based on a high confidence score. However, the task fails as not all cupboards contain a blue can. In contrast, with the ASR method (bottom row), the agent detects a discrepancy in the object-object relationship, recognizing that a blue can is already placed in the right cupboard. The agent then moves the left blue can to the empty cupboard, successfully completing the task.

**No PRED.** Finally, we conduct an ablation study when all the methods are excluded ((a) *vs.* (f)). We observe a substantial performance gap across all splits including valid unseen in TEACh and test unseen in ALFRED, which demonstrates that PRED greatly contributes to solving the tasks.

Table 3: **Ablation study in TEACh and ALFRED for each proposed component.** The path-length-weighted (PLW) metrics are given in the parentheses for each value. (b) to (e) show the performances of PRED without each component.

| Model | TfD | | EDH | | ALFRED | |
|---|---|---|---|---|---|---|
| | Valid Unseen | | | | Test Unseen | |
| | SR | GC | SR | GC | SR | GC |
| (a) **PRED** | 19.77 (5.16) | 16.74 (8.31) | 21.69 (4.44) | 26.83 (7.46) | 32.31 (12.48) | 42.62 (17.87) |
| (b) w/o DTA | 17.65 (4.63) | 13.39 (6.63) | 20.67 (3.52) | 25.74 (6.62) | 29.63 (11.79) | 39.49 (16.66) |
| (c) w/o OHV | 15.20 (4.22) | 12.58 (6.57) | 20.35 (3.44) | 22.89 (6.66) | 29.30 (11.33) | 38.95 (16.54) |
| (d) w/o APM | 16.18 (4.14) | 13.66 (6.45) | 19.89 (3.09) | 24.93 (5.85) | 24.33 (9.13) | 32.11 (13.98) |
| (e) w/o ASR | 17.16 (4.44) | 15.30 (7.41) | 20.49 (4.06) | 26.00 (7.42) | 28.25 (11.22) | 38.98 (17.52) |
| (f) w/o all | 12.91 (2.91) | 10.86 (5.62) | 18.65 (3.09) | 19.06 (5.80) | 23.35 (8.72) | 31.25 (14.61) |

## 4.3 Qualitative Analysis

As shown in the top row of Figure 3 (without ASR), the task fails despite completing the initial plan. In contrast, as seen in the bottom row with ASR, considering object-object relationships enables faster and more successful task completion. We also provide qualitative analyzes of DTA, OHV, APM, and ASR including simulated environments and real robot experiments in Appendix G.

## 5 Conclusion

We propose PRED that revises their behaviors based on perceived environmental discrepancies inspired by brain plasticity of humans and animals before failure. Given perceived environmental discrepancies, PRED builds prompts, and queries LLMs to generate a revised plan. To address these environmental discrepancies, we propose DTA, OHV, APM, and ASR for replanning for effective and efficient task completion. We observe that our PRED outperforms previous methods notably in two challenging embodied instruction following benchmarks, TEACh and ALFRED.

**Limitations and future work.** The environmental discrepancies are perceived based on semantic information (*e.g.*, object masks, semantic spatial maps, *etc*) predicted from a single egocentric observation and therefore may not be accurate, possibly leading to inaccurate plan modification. A promising future direction is to modify a plan even with these possibly inaccurate discrepancies.

**Acknowledgments**

This work was partly supported by the IITP grants (No.RS-2022-II220077, No.RS-2022-II220113, No.RS-2022-II220959, No.RS-2022-II220871, No.RS-2020-II201361 (Yonsei AI), No.RS-2021-II211343 (SNU AI), No.RS-2021-II212068 (AI Innov. Hub), No.RS-2022-II220951) funded by the Korea government (MSIT). Authors thank to Seungmin Kim and Prof. Sungjoon Choi at Korea University for helping us about real robot experiments.

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

**Note:** Orange denotes reference to the main paper.

## A Additional Details of Related Work

Many recent researches try to use large language models (LLMs) to make their plan and revise it. For example, [30] retrieves the most relevant top-K examples of error correction, while [45] condenses a robot's previous experiences to identify failures and devises new plans. [47] plans a sequence of subgoals and revises its plan upon failure.

[54] mitigates this by reasoning and proposing the next subgoal after completing the previous one, implicitly considering failure in predicting the next step, but it still does not account for failures during subgoal execution.

[48] proposes correcting the error or preventing the failure using LLMs with predefined preconditions. [49] proposes to generate complete plans using a structured programming language prompt, with each action considering conditions for error-free execution.

[50,51,52] uses LLMs for self-corrective ability by receiving environmental feedback, but depends on ground-truth feedback from environments associated with a small number of actions. Recent approaches [53, 30] sometimes exploit ground-truth information (*e.g.*, detection results, oracle information such as, "what causes task failure?," *etc*) for replanning. For instance, a recent approach [53] exploits either ground truth or expensive human feedback.

[30] updates plans based on full RGB egocentric observations (i.e., images), which could make the model sensitive to irrelevant factors like background changes. [46] adjusts plans using detected objects from the current observation as feedback after navigation failure but overlooks visual details such as appearance, attributes, and relationships.

[47,54] limitedly address environmental discrepancies. They identify whether a target object is at the expected destination and propose new target candidates if it is not (*i.e.*, object presence discrepancy). However, they do not address the other discrepancies (*i.e.*, appearance, attribute, and relationship discrepancies), potentially reducing their effectiveness and efficiency.

## B Additional Details of Agent Architecture

Inspired by the recent success of the SLAM-based approaches [42, 44, 18], we adopt an architecture that 1) plans a sequence of subgoal actions from a natural language instruction, 2) maintains the agent's memory in the form of a semantic spatial map built by an observation history with depth and masks predicted by pretrained perception models [60], [44], and 3) uses a deterministic algorithm [43] over the semantic spatial map for effective obstacle-free path planning. Inspired by the recent success of the SLAM-based approaches [42, 44, 18], we adopt an architecture that 1) plans a sequence of subgoal actions from a natural language instruction, 2) maintains the agent's memory in the form of a semantic spatial map built by an observation history with depth and masks predicted by pretrained perception models [60], [44], and 3) uses a deterministic algorithm [43] over the semantic spatial map for effective obstacle-free path planning.

**Semantic spatial map.** During exploration, the agent takes as input the current egocentric RGB observation, $v_t$, for each time step, $t$, and predicts its semantic information (*e.g.*, object masks, $\{m_i\}_{i=0}^N$, a depth map, $d_t$, a spatial semantic map, $Z_t$, *etc*). Using $\{m_i\}_{i=0}^N$, the agent obtains the cropped images of objects, $\{v_i^c\}_{i=0}^N$, from $v_t$. Here, $N$ denotes the number of all detected objects in $v_t$, and $v_i^c$ an object's image crop with the masked-out background. $Z_t$ denotes a 2D top-down map predicted by merging three inputs and is generated by the semantic map generator, $\mathcal{G}$, as follows:

$$Z_t = \mathcal{G}(d_t, \{m_i\}_{i=0}^N, Z_{t-1}). \tag{2}$$

$G$ transforms each pixel of $d_t$ and $\{m_i\}_{i=0}^N$ into a 3D point with its semantic label. These points are then summed along the gravitational axis to generate the current top-down semantic spatial map

containing only current information. Finally, $G$ accumulate the obtained top-down map on $Z_{t-1}$, resulting in an updated semantic spatial map, $Z_t$.

**Navigation policy.**    To interact with an object, an agent must first reach the object in its close vicinity. For this, previous approaches [37] often use behavior cloning [61] to let the agent mimic the navigational behavior of a vision-language navigation expert. However, such behavior cloning requires a large number of training trajectories and natural language annotations for satisfactory performance, but collecting these may not be trivial due to high computational costs and time.

To address this issue, recent approaches [41, 42, 26, 30] instead incorporate deterministic algorithms (*e.g.*, A*, FMM [43], *etc*) to plan obstacle-free paths and observe significant improvement of navigational performance while alleviating the data collection burden. Inspired by this improvement, we also adopt a deterministic policy [43] to plan navigation routes.

## C    Additional Details of PRED

We provide more details of PRED. We further detail how we revise an action plan using large language models (LLMs) based on perceived environmental feedback, illustrated in Figure 4.

Additionally, we explain the 'Appearance Detector Module' of OHV, which takes egocentric views in different views as input and predicts the class of the picked-up object; the 'Attribute Detector Module' of APM, which takes the cropped image as input and predicts the object's attributes; and the 'Relationship Detector Module' of ASR, which takes the predicted masks as input and predicts the relationship between the target object and where it can be placed.

### C.1    Revising Actions by Environmental Feedback with LLMs

**Subgoal planning.**    Given a natural language instruction, $\mathcal{X}$, such as dialogs or directives, a large language model, $\mathcal{L}$, predicts the task-relevant contexts, $\mathcal{E}$, such as task type, target object, its expected location, *etc*. Then, they are integrated into an LLM-generated high-level action plan, $\mathcal{T}$, for the corresponding task type through the integration process, 'Intg.'

Here, $\mathcal{T}$ is created to satisfy the desired states of the task type. For example, if the task type is "CLEAN ALL X" and X is an object, the desired state is defined as object = {clean}. To create the plan, we assume that the initial state is the opposite of the desired one, such as object = {dirty}, and then create a plan to clean the object.

Integration of $\mathcal{T}$ and $\mathcal{E}$ resembles context-aware planning [18], but we use an unfine-tuned LLM for context prediction instead of a trained model with benchmark data. Specific information obtained from the instruction can be incorporated into the action plan. For example, if the instruction indicates the mug is inside the fridge, an action to open the fridge is added before picking up the mug.

The generated action plan consisting of high-level actions, $\{a_n^h\}_{n=1}^{N_h}$, in a triplet format is systematically converted to the executable action plan with low-level actions, $\{a_n^l\}_{n=1}^{N_l}$, in a tuple format by a rule-based function, $f_c$, following the prior work [18]. This process is expressed as in Equation 3:

$$\mathcal{E} = \mathcal{L}(\mathcal{X}), \quad \{a_{n_h}^h\}_{n_h=1}^{N_h} = \text{Intg}(\mathcal{T}, \mathcal{E}), \quad \{a_{n_l}^l\}_{n_l=1}^{N_l} = f_c(\{a_{n_h}^h\}_{n_h=1}^{N_h}). \tag{3}$$

**Revising actions.**    While performing the task, the agent may encounter unexpected environmental discrepancies (see Section 3.1). To translate the discrepancies detected by each module into natural language, we use an LLM to generate a feedback prompt that describes the discrepancies. We provide the LLM with the Python codes corresponding to the detecting discrepancy and a prompt explaining its role. Then, LLM generates the feedback prompt such as "The target is either directly visible to the agent or its information is already stored on the map."

Once we obtain environmental feedback (*i.e.*, environmental discrepancy) based on visual input from the LLM, we build a prompt, $\mathcal{P}$ with this feedback to query $\mathcal{L}$ for plan revision. Specifically, we build the prompt with 1) a system prompt, $\mathcal{P}_s$, 2) the original plan (*i.e.*, current action plan in Figure 4), $\{a_n\}_{n=1}^{N}$ ($\{a_n^h\}_{n=1}^{N_h}$ or $\{a_n^l\}_{n=1}^{N_l}$), and 3) the feedback prompt, $\mathcal{P}_f$.

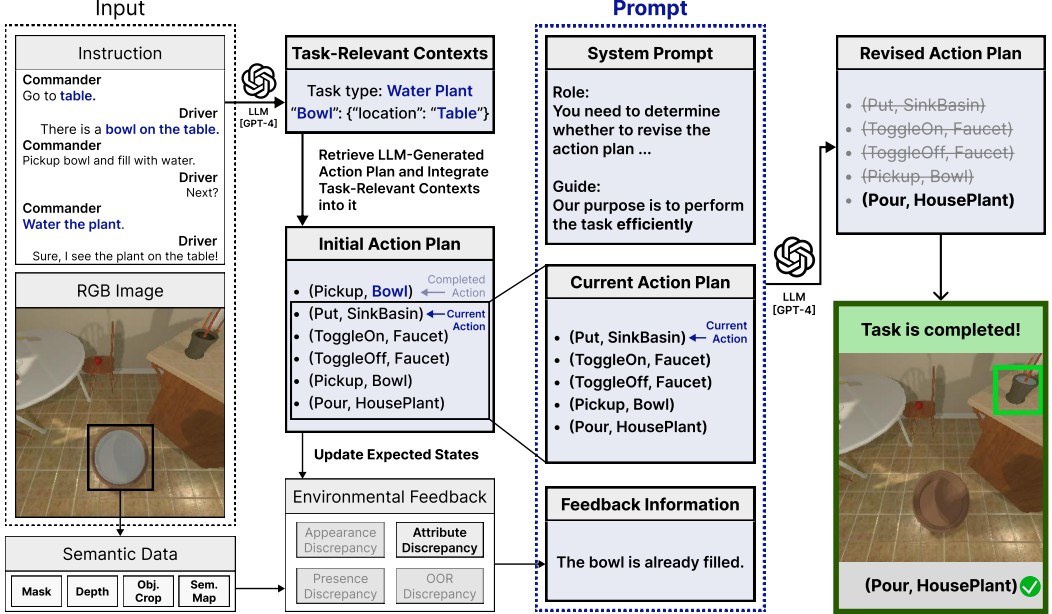

Figure 4: **Overall Process of Revising Actions by LLMs.** We first extract task-relevant contexts from the dialog using an LLM. These contexts are integrated into a retrieved action plan generated by the LLM to form an initial action plan. Using semantic data from an RGB image and expected states from the initial plan, PRED verifies discrepancies as environmental feedback. The prompt including the system prompt explaining the role and guide for the LLM, the current action, and the feedback information is given to the LLM to generate a revised action plan. The bottom right image shows the task is completed along with the efficiently revised plan.

The format of the current action plan, $\{a_n\}_{n=1}^{N}$, can be the sequence of the actions either in an executable tuple (low-level action), $a_{n_l}^{l}$, or a triplet (high-level action), $a_{n_h}^{h}$. A triplet action is predefined and consists of a unit of action that includes several executable actions. For example, ('Move', 'Target', 'Parent') signifies the combination of ('Target', 'PickupObject') and ('Parent', 'PutObject'). We operate with both types of action plans simultaneously. For instance, the action plan is input in triplet form if the actions can be revised at the triplet level, while in tuple form if the revision is required at the detailed executable action level. If the output is in triplet form, it is systematically translated into executable tuple form by $f_c$. This revision process continues whenever the environmental discrepancy is detected until the agent successfully completes the task.

$\mathcal{P}$ for $\mathcal{L}$ varies according to the type of discrepancy and given action plan consisting of various objects and actions. We provide several examples of the $\mathcal{P}$ including $\mathcal{P}_s$, a current action plan, $\mathcal{P}_f$, and the corresponding output of the $\mathcal{L}$ in Listing 1 to 5. We provide the prompt to extract $\mathcal{E}$ in Listing 6 and one to generate a revised action plan in Listing 7.

## C.2 Appearance Detector Module

The agent often fails at object interaction due to misperception caused by lighting, occlusion, *etc*. To address this issue, we propose to verify that an interacted object is the intended one. The intuition is that correct perception will yield consistent perception results.

Specifically, we verify that two predicted classes, $c_0$ and $c_i$, of the target object, $o$ are identical. Here, $c_0$ is the predicted class of $o$ from the $0^{th}$ viewpoint at the time of interaction, and $c_i$ is that of $o$ from the $i^{th}$ viewpoint obtained from a different pose from the one where the $0^{th}$ viewpoint is obtained. The agent predicts $c_0$ and $c_i$ from egocentric images of the object at the $0^{th}$ and $i^{th}$ viewpoints with a pretrained perception model. If $c_0 \neq c_i$, the agent revises its actions accordingly.

There are various ways to obtain different viewpoints of an object, such as grasping it from different directions or rotating it. Here, we simply lift or lower the interacted object implemented by appropriate actions (here, LOOKUP and LOOKDOWN in the Appendix D) supported by the agent's embodiment.

A better method of obtaining these viewpoints may improve our proposed module, but this is beyond our scope and thus, we leave this as future work.

### C.3 Attribute Detector Module

We consider attributes related to the desired state to determine the success of the task. Furthermore, the expected attribute is one of the initial states related to the attribute, and the initial states are defined as the opposite of the desired state (detailed in the Appendix C.1).

We take a retrieval-based approach to predict the attribute, $\hat{\phi}_o$, of a detected target object, $o$. This approach is preferred over training a model because it requires neither extensive training nor large datasets. Additionally, if a new attribute is added, a training-based model must be retrained or fine-tuned, whereas the retrieval-based approach only needs to add new options for comparison.

To predict $\hat{\phi}_o$, we retrieve the most similar image and assign the attribute of the object in the retrieved image to the detected target. To compare images focusing only on the object and excluding the background, we use the cropped image of the target object, $v_o^c$, and the cropped images, $\{v_i^{ct}\}_{i=0}^N$, from the training dataset. We compare these cropped images through cosine similarity, 'Cos', after extracting features processed by a ViT-B/32 model [62], pre-trained with CLIP weights [2], as:

$$\operatorname*{argmax}_i \operatorname{Cos}(v_o'^c, v_i'^{ct}) \quad i = 0, 1, \ldots, N, \tag{4}$$

where $v_o'^c$ and $v_i'^{ct}$ are the features of $v_o^c$ and $v_i^{ct}$, respectively. Then, we consider $\hat{\phi}_o$ as the attribute of the object in $v_i'^{ct}$ having the highest cosine similarity score.

### C.4 Relationship Detector Module

To determine the relationship between the target object, $o$, and the object it is placed on, $o_p$, we need to find the most 'adjacent' mask, $m_i$, of $o_p$ to the target object's mask in all detected masks, $\{m_i\}_{i=0}^N$, in the current egocentric view. Here, the mask at $i = 0$ is defined as the mask of the target object, $m_0$, of $o$. To find $m_i$, we dilate $\{m_i\}_{i=0}^N$ and calculate the intersection over union (IoU) between the enlarged mask of the target object, $m_0'$, and the enlarged masks of other objects, $\{m_i'\}_{i=1}^N$ as follows:

$$\operatorname*{argmax}_i \operatorname{IoU}(m_0', m_i'); \quad i = 1, 2, \ldots, N. \tag{5}$$

We define $m_i'$ as the mask with the largest IoU score with $m_0'$, considering it the 'adjacent' mask. The object represented by $m_i'$ is referred to as the object $o_p$ that the target object is placed on.

## D Benchmark and Baseline Details

We validate PRED in two challenging benchmarks: TEACh [13], for dialog instruction following, and ALFRED [12], which provides declarative instructions, to assess generalization in different task setup. We provide details for each benchmark and baselines used below.

### D.1 TEACh

**Benchmark.** The TEACh benchmark aims to allow agents to navigate and interact with objects based on instructions, with task completion achieved by meeting specified conditions, such as cleaning at least one mug for the instruction "clean a mug".

The instruction is a dialog in natural language, which is comprised of two components: the COM-MANDER that provides task-relevant information based on oracle information about the task and the FOLLOWER that performs the task through the dialog. Upon receiving instructions, the FOLLOWER translates the natural language instructions and egocentric visual observations into executable actions. The executable actions are expected to succeed in the task.

The agent can take 16 different actions. Eight actions (FORWARD, BACKWARD, TURN LEFT, TURN RIGHT, LOOK UP, LOOK DOWN, STRAFE LEFT, STRAFE RIGHT) are designated for navigation,

and the other eight actions (PICKUP, PLACE, OPEN, CLOSE, TOGGLEON, TOGGLEOFF, SLICE, AND POUR) are for interaction. Navigation actions are discrete: head movements are by 30°, turns are by 90°, and movements are in 0.25m increments. During interaction, the agent selects the object at coordinate (x, y) in its egocentric view.

Additionally, the TEACh benchmark focuses on Execution from Dialogue History (EDH) and Trajectory from Dialogue (TfD). This benchmark is divided into train, validation, and test splits. Evaluation metrics encompass success rate (SR), goal-condition success rate (GC), and path-length-weighted (PLW) scores.

**State-of-the-art baseline models.**   We compare our PRED with the recently proposed state-of-the-art methods: E.T. [37], JARVIS [41], FILM [42], DANLI [26], and HELPER [30]. E.T. learns a direct mapping from a natural language dialog and an egocentric observation to a corresponding action and the position of an object to be interacted with. JARVIS employs an LLM trained on the TEACh dialog dataset to produce high-level subgoals, replicating the ones executed by human demonstrators. It utilizes a semantic map alongside the E.T. to locate objects. FILM enhances an LLM through fine-tuning to generate a parameterized plan. Mirroring Jarvis, it leverages a semantic map to execute subgoals and employs a semantic policy for object search. DANLI fine-tunes an LLM for high-level subgoal prediction and employs symbolic planning with an object state and spatial map for execution plan formulation. It incorporates an object search module and manual error correction mechanisms. HELPER utilizes a Large Language Model (LLM) to generate initial high-level actions with additional data. When a failure occurs, it predicts the error reasons through a pretrained vision and language model, and revises the action using the LLM.

## D.2   ALFRED

**Benchmark.**   The ALFRED benchmark requires agents to complete a long-horizon task by understanding declarative natural language instructions with egocentric observations. The declarative instructions comprise two types of instruction: one is a high-level description that provides a single sentence to complete the task and the other is a step-by-step instruction that details the process of performing a task. By following the instructions, the agent executes the two types of predefined actions. The navigation actions include MOVEAHEAD, ROTATERIGHT, ROTATELEFT, LOOKUP, and LOOKDOWN. The interaction actions include PICKUPOBJECT, PUTOBJECT, OPENOBJECT, CLOSEOBJECT, TOGGLEOBJECTON, TOGGLEOBJECTOFF, and SLICEOBJECT.

**State-of-the-art baseline models.**   We compare our PRED with the recently proposed state-of-the-art methods: HLSM [44], FILM [57], and CAPEAM [18]. HLSM employs a hierarchical controller to translate natural language instructions into actions the agent can execute. The high-level controller identifies the next subgoal based on the given instructions and map, while the low-level controller generates a sequence of actions to accomplish this subgoal. FILM uses a pre-constructed template as a high-level action plan. It employs two BERT [6] classifier submodules to identify the instruction type and determine the template arguments. It applies a deterministic algorithm [43] to plan a path without obstacles. CAPEAM employs context-aware planning to devise a sequence of subgoals and execute each subgoal using the appropriate detailed planners. It also utilizes extra memory to avoid interacting with unsuitable objects.

# E   Additional Experiment

## E.1   Real Robot Experiments

We conduct experiments in a real environment to evaluate the effectiveness of our proposed method. To perceive the environment and obtain feedback, we use grounded SAM [63]. Additionally, we employed the UR5 manipulator for object interaction. First, we recognize the objects and environment from an egocentric view using a camera mounted on the back of the manipulator's gripper.

Table 4: **Alternative TEACh EDH evaluation split.** The path-length-weighted (PLW) metrics are given in the parentheses for each value. The highest and second highest values per fold and metric are shown in **bold** and underline, respectively.

| Model | Validation | | | | Test | | | |
|---|---|---|---|---|---|---|---|---|
| | Unseen | | Seen | | Unseen | | Seen | |
| E.T. [37] | 8.35 (0.86) | 6.34 (3.69) | 8.28 (1.13) | 8.72 (3.82) | 7.38 (0.97) | 6.06 (3.17) | 8.82 (0.29) | 9.46 (3.03) |
| DANLI [26] | 17.25 (7.16) | 23.88 (19.38) | 16.89 (9.12) | 25.10 (22.56) | 16.71 (7.33) | 23.00 (20.55) | 18.63 (9.41) | 24.77 (21.90) |
| HELPER [30] | 17.25 (3.22) | 25.24 (8.12) | 19.21 (4.72) | 33.54 (10.95) | 17.55 (2.59) | 26.49 (7.67) | 17.97 (3.44) | 30.81 (8.93) |
| **PRED** (Ours) | **21.52**(4.64) | **26.88**(7.25) | **23.84**(4.20) | **33.79**(10.64) | **22.04**(4.24) | **26.77**(7.67) | **19.61**(4.96) | **31.86**(10.19) |

Table 5: **Ablation study in TEACh for each proposed component.** The path-length-weighted (PLW) metrics are given in the parentheses for each value. (b) to (e) show the performances of PRED without each component.

| Model | TfD | | | | EDH | | | |
|---|---|---|---|---|---|---|---|---|
| | Unseen | | Seen | | Unseen | | Seen | |
| | SR | GC | SR | GC | SR | GC | SR | GC |
| (a) **PRED** | 19.77 (5.16) | 16.74 (8.31) | 20.99 (4.64) | 21.55 (11.03) | 21.69 (4.44) | 26.83 (7.46) | 21.71 (4.62) | 32.78 (10.39) |
| (b) w/o DTA | 17.65 (4.63) | 13.39 (6.63) | 16.02 (4.05) | 16.34 (8.13) | 20.67 (3.52) | 25.74 (6.62) | 18.75 (3.59) | 30.95 (8.69) |
| (c) w/o OHV | 15.20 (4.22) | 12.58 (6.57) | 16.02 (4.56) | 17.46 (9.68) | 20.35 (3.44) | 22.89 (6.66) | 18.75 (3.87) | 29.87 (9.60) |
| (d) w/o APM | 16.18 (4.14) | 13.66 (6.45) | 14.36 (3.70) | 18.59 (9.10) | 19.89 (3.09) | 24.93 (5.85) | 19.98 (3.67) | 29.42 (7.40) |
| (e) w/o ASR | 17.16 (4.44) | 15.30 (7.41) | 18.23 (5.11) | 17.18 (8.92) | 20.49 (4.06) | 26.00 (7.42) | 18.91 (2.53) | 30.62 (9.38) |
| (f) w/o all | 12.91 (2.91) | 10.86 (5.62) | 12.15 (3.31) | 15.49 (7.41) | 18.65 (3.09) | 19.06 (5.8) | 17.95 (3.77) | 24.39 (8.16) |

Once the agent identifies the desired object using [63] and determines which object to interact with, it retrieves the detected mask of the object as input to select the correct object in the real world. The agent then receives the ground truth position of the selected object and proceeds with the interaction.

### E.2 Results on New Validation Split

We conduct an additional experiment, exclusively done by DANLI [26] for fair comparison, in the TEACh benchmark [13]. In this benchmark, we investigate the performance of our PRED in different splits provided on the TEACh GitHub and in [26]. The leaderboard for EDH of the TEACh benchmark is unavailable, preventing the evaluation on its true test set. Thus, we leveraged the original validation

Table 6: **Ablation study in ALFRED for each proposed component.** The path-length-weighted (PLW) metrics are given in the parentheses for each value. (b) to (e) show the performances of PRED without each component.

| Model | Test Unseen | | Test Seen | |
|---|---|---|---|---|
| | SR | GC | SR | GC |
| (a) **PRED** | 32.31 (12.48) | 42.62 (17.87) | 35.09 (15.02) | 43.27 (19.95) |
| (b) w/o DTA | 29.63 (11.79) | 39.49 (16.66) | 33.59 (13.92) | 42.02 (18.69) |
| (c) w/o OHV | 29.30 (11.33) | 38.95 (16.54) | 33.53 (13.91) | 42.75 (19.07) |
| (d) w/o APM | 24.33 (9.13) | 32.11 (13.98) | 24.66 (10.01) | 32.42 (15.00) |
| (e) w/o ASR | 28.25 (11.22) | 38.98 (17.52) | 33.07 (14.48) | 42.95 (19.63) |
| (f) w/o all | 23.35 (8.72) | 31.25 (14.61) | 24.27 (9.95) | 32.24 (15.49) |

splits for seen/unseen scenarios, aligning with the approach taken in most prior studies [30, 41, 42].

In Table 4, we present the alternative validation and test splits. We observe that our method outperforms others in the new EDH split, achieving improvements with notable margins in SR and GC, similar to the results in the original split. In the seen environment, the agent encounters fewer misperceptions and navigation errors than in the unseen environment, making our PRED less effective.

## F  Additional Ablation Study

We provide the results of ablation studies in all splits including 'Seen' environments in TEACh and ALFRED benchmark in Table 5 and 6.

## G  Qualitative Analysis

**DTA.** First, Figure 5a describes two different simulation scenarios: PRED without (top) and with (bottom) DTA in Section 3.1.1. An agent generates an initial plan considering the information mentioned in the instruction ("mug, potentially in a cabinet") so that it predicts that the mug is in the cabinet and has a plan of opening the cabinet first, instead of picking up the mug. Thus, in the upper

scenario (w/o DTA), the agent keeps its original plan to open the cabinet even if the usable mug is observed in its sight. On the contrary, the agent with DTA changes the plan to skip the action with the cabinet and to pick up the mug right away after perceiving the mug. As a result, it can finish the task efficiently as it adapts to the discrepancy of the target object's presence in the environment.

Figure 6a shows two real robot scenarios: one without DTA (top) and one with DTA (bottom). The agent forms an initial plan based on the instruction ("Pepsi, possibly in a red cup"), assuming the Pepsi is in the red cup and planning to move the cup before picking up the Pepsi. In the scenario without DTA (top), the agent follows the initial plan and moves the red cup, even though another Pepsi is visible. In contrast, with DTA (bottom), the agent adjusts the plan, skipping the red cup and picking up the Pepsi immediately after detecting it. This adaptation allows the agent to complete the task more efficiently by adapting to the actual environment.

**OHV.** Figure 5b illustrates the difference with and without OHV in the simulation, as described in Section 3.1.2. The first image shows the agent needing to pick up a 'Mug' but mistakenly picking up a 'Cup' instead by misperception. The agents without OHV will not try to correct the wrong action since they do not consider the appearance discrepancy after the action is done. In contrast, our agent with OHV verifies whether the interacted object aligns with the desired one by examining its appearance from multiple angles. If it detects the discrepancy, the agent rectifies its mistake by revising actions. As a result, OHV helps the agent to prevent the failure that comes from the interaction with the wrong object.

Figure 6b demonstrates the impact of OHV in real robot experiments. At t = 1, with the scenarios with OHV (top row) and without OHV (bottom row), the agents experience misperception because the color and design of the cans look almost identical from the top-down view. As a result, at t=2, the agent picks the can with the higher confidence score but mistakenly grabs a drink other than the desired Fanta. In the scenario without OHV (top row), the agent sticks to the original plan, ignoring the mistake, and delivers the wrong item to the destination. However, with OHV (bottom row), the agent views the drink from the front, identifies the label, and realizes it has picked the wrong item, detecting the appearance discrepancy. The agent then puts down the incorrect item, updates the plan, and successfully completes the task.

**APM.** Next, we investigate the benefit of APM (in Section 3.1.3) depicted in Figure 5c with the scenarios where agents want to open a microwave. The agent without APM attempts to open it as it supposes that the microwave is not operating, leading to an interaction failure. In contrast, the agent with APM considers the attribute discrepancy coming from the difference in the microwave's state. With this, it adds actions that toggle it off first to future actions, enabling successful interaction.

**ASR.** Finally, we elucidate the advantage of ASR in Section 3.1.4. Figure 5d describes the benefit of considering relationship discrepancy. We refer to the relationship corresponding to the goal condition in the instruction as the goal relationship. An agent without ASR may pick up a remote and put it down with a higher confidence score, even though it already satisfies the goal relationship, resulting in only one remote on the table. If this were the second interaction (picking it up and putting it down on the table), the agent would think that it completed the task since it completed interactions twice. However, the task would fail since there are no two remotes on the table. In contrast, the agent with ASR considers relationship discrepancy and interacts with a remote not in the goal relationship instead of the one already in a goal relationship.

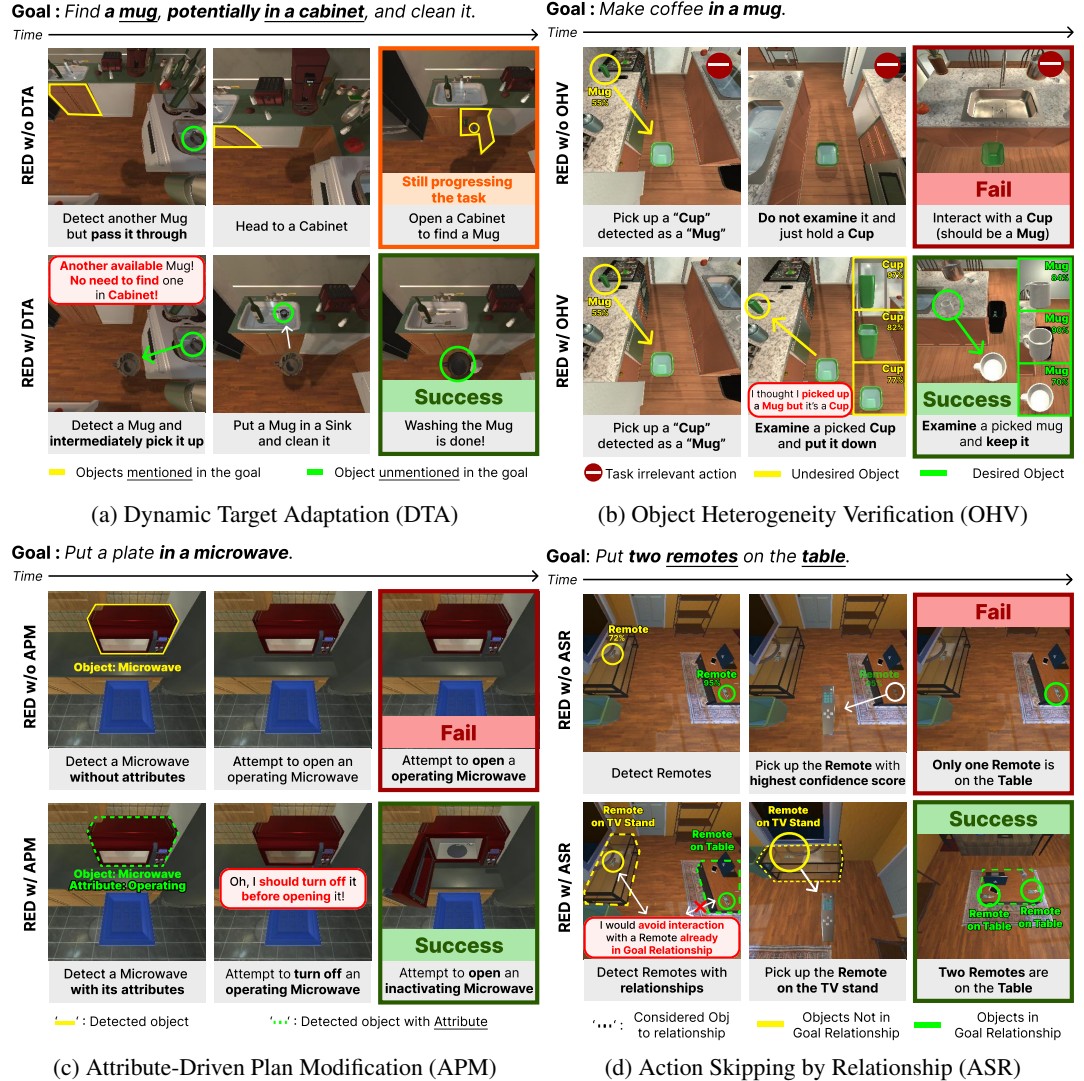

(a) Dynamic Target Adaptation (DTA)

(b) Object Heterogeneity Verification (OHV)

(c) Attribute-Driven Plan Modification (APM)

(d) Action Skipping by Relationship (ASR)

Figure 5: Qualitative analysis of benefit of DTA, OHV, APM, and ASR.

**Goal :** Place a **Pepsi**, possibly in a **red cup**, on the **yellow cupboard**
**Initial Plan :** (Pickup, RedCup), (Put, Table), (Pickup, Pepsi), (Put, YellowCupboard)

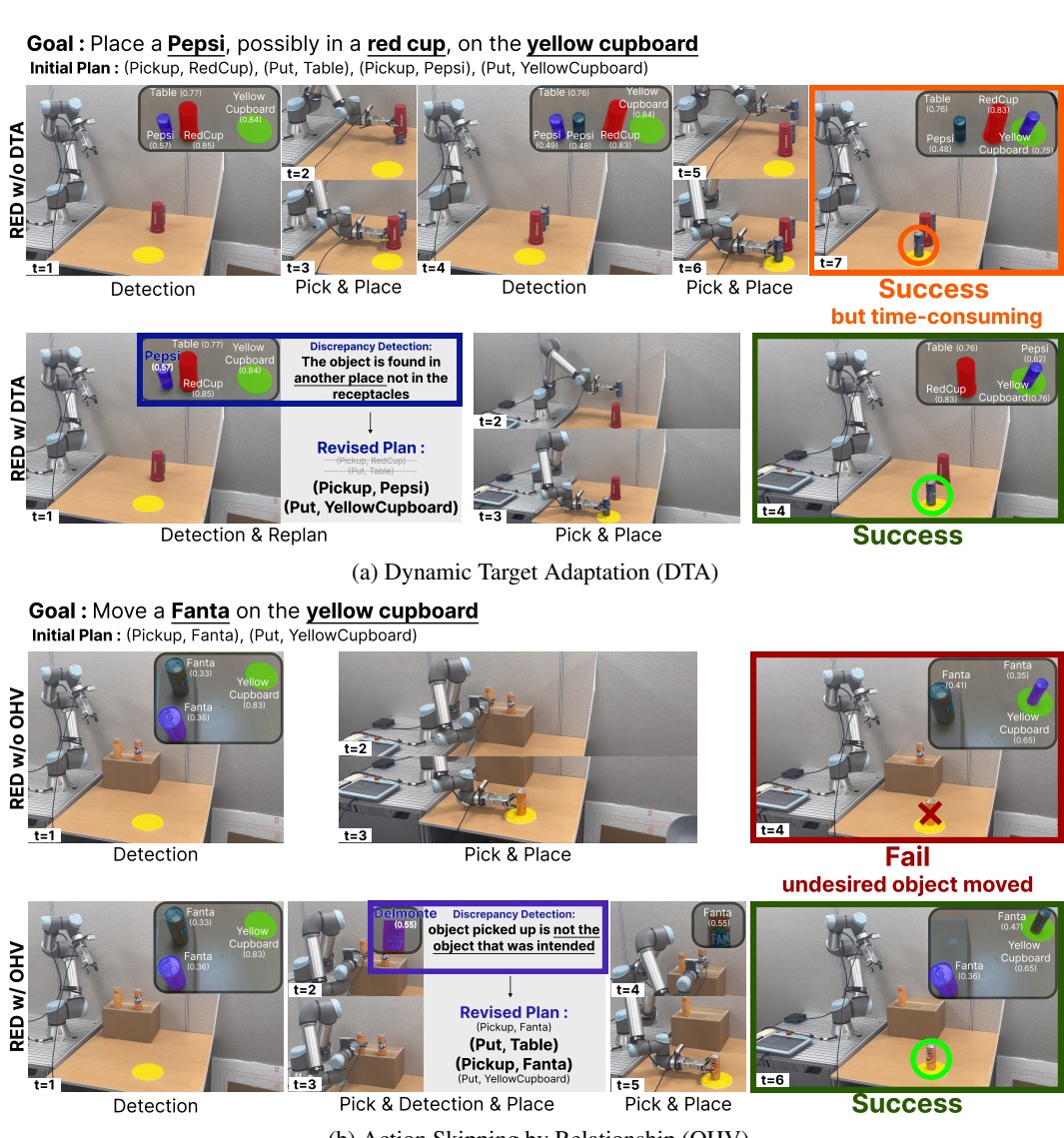

(a) Dynamic Target Adaptation (DTA)

**Goal :** Move a **Fanta** on the **yellow cupboard**
**Initial Plan :** (Pickup, Fanta), (Put, YellowCupboard)

(b) Action Skipping by Relationship (OHV)

Figure 6: Qualitative analysis of benefit of DTA and OHV

```
### INPUT

# System Prompt

You need to determine whether to revise the action sequence to solve
    the task considering 'INFO' and then write down the final sequence
     of actions if needed.
This action sequence is made for solving household task. Each action's
     format is tuple.
The possible action spaces are as follows.
['Target', 'PickupObject'], ['Target', 'PutObject'], ['Target', '
    OpenObject'], ['Target', 'CloseObject'], ['Target', '
    ToggleObjectOn'], ['Target', 'ToggleObjectOff'], ['Target', '
    PourObject'], ['Target', 'SliceObject']
For example, ['Target', 'PickupObject'] is a command to pick up the
    target.
Modify the action sequence by adding or subtracting action to suit the
     situation if needed. If it is not needed to be revised, just
    write down given action sequence.
In this context, 'action' refers to each element in tuple format in
    the list(action sequence).
You can add or subtract an action at the very first part of the action
     sequence if needed. Do not modify the actions that follow, and
    write them down as originally provided.
Do not add additional explannation. Just write the final action
    sequence in the right format (tuples in list).

GUIDE: When you want to pick up an object, it sometimes is located in
    receptacles that should be opened to pick up the object.
In this case, action sequence contains the sequence of open, pickup(or
     slice), close.
Your objective is just picking up(or slicing) the object.

# Current Action Plan

        [
            "Recep",
            "OpenObject"
        ],
        [
            "Target",
            "PickupObject"
        ],
        [
            "Recep",
            "CloseObject"
        ],
        [
            "Apple",
            "SliceObject"
        ],
        [
            "CounterTop",
            "PutObject"
        ]

# Feedback Information

The target is either directly visible to the agent or its information
    is already stored on the map.

### OUTPUT

        [
```

```
            "Target",
            "PickupObject"
        ],
        [
            "Apple",
            "SliceObject"
        ],
        [
            "CounterTop",
            "PutObject"
        ]
```

Listing 1: **Example of the input(*i.e.*, Prompt) and the output (*i.e.*, Revised Action Plan) of LLM.** This example describes the result where DTA is applied. Compared to the current action plan, the action plan is revised to skip the actions (*i.e.*, open and close the receptacle) to make a plan efficient. Target and Recep will be replaced with each corresponding object based on the context.

```
### INPUT

# System Prompt

You need to determine whether to revise the action sequence to solve
    the task considering 'INFO' and then write down the final sequence
     of actions if needed.
This action sequence is made for solving household task. Each action's
     format is tuple.
The possible action spaces are as follows.
['Target', 'PickupObject'], ['Target', 'PutObject'], ['Target', '
    OpenObject'], ['Target', 'CloseObject'], ['Target', '
    ToggleObjectOn'], ['Target', 'ToggleObjectOff'], ['Target', '
    PourObject'], ['Target', 'SliceObject']
For example, ['Target', 'PickupObject'] is a command to pick up the
    target.
Modify the action sequence by adding or subtracting action to suit the
     situation if needed. If it is not needed to be revised, just
    write down given action sequence.
In this context, 'action' refers to each element in tuple format in
    the list(action sequence).
You can add or subtract an action at the very first part of the action
     sequence if needed. Do not modify the actions that follow, and
    write them down as originally provided.
Do not add additional explannation. Just write the final action
    sequence in the right format (tuples in list).

GUIDE: When picking up an object, it may not always be the intended
    object.
But you can only handle one object in your hand which means you have
    to put the object and repick the object if you want to pick
    another object.
When you are not sure where to put the object, just use 'Parent' as
    the place you put it on. (In the action, ['A', 'PutObject'], 'A'
    should be the location to place the object on, not the object that
     you are holding.)
Let's assume that you have done the first action.
In this case, do not revise the first action, but if you think some
    actions should be added, do it right after the first action.

# Current Action Plan

        [
            "Target",
            "PickupObject"
        ],
        [
            "Bed",
            "PutObject"
        ]

# Feedback Information

After checking, the picked up object's mask is detected, but the
    object is not the desired one as its center coordinates do not
    fall within the specified range (width between 100 and 200, height
     between 100 and 230).

### Output

        [
            "Target",
            "PickupObject"
```

```
        ],
        [
            "Parent",
            "PutObject"
        ],
        [
            "Target",
            "PickupObject"
        ],
        [
            "Bed",
            "PutObject"
        ]
```

Listing 2: **Example of the input(*i.e.*, Prompt) and the output (*i.e.*, Revised Action Plan) of LLM.** This example describes the result where OHV is applied. Compared to the current action plan, the actions (*i.e.*, put the target on the parent and pick up the target again) are inserted to put down the wrong object and pick up an appropriate one. Target and Parent will be replaced with each corresponding object based on the context.

```
### INPUT

# System Prompt

You need to determine whether to revise the action sequence to solve
    the task considering 'INFO' and then write down the final sequence
     of actions if needed.
This action sequence is made for solving household task. Each action's
     format is tuple.
The possible action spaces are as follows.
['Target', 'PickupObject'], ['Target', 'PutObject'], ['Target', '
    OpenObject'], ['Target', 'CloseObject'], ['Target', '
    ToggleObjectOn'], ['Target', 'ToggleObjectOff'], ['Target', '
    PourObject'], ['Target', 'SliceObject']
For example, ['Target', 'PickupObject'] is a command to pick up the
    target.
Modify the action sequence by adding or subtracting action to suit the
     situation if needed. If it is not needed to be revised, just
    write down given action sequence.
In this context, 'action' refers to each element in tuple format in
    the list(action sequence).
You can add or subtract an action at the very first part of the action
     sequence if needed. Do not modify the actions that follow, and
    write them down as originally provided.
Do not add additional explannation. Just write the final action
    sequence in the right format (tuples in list).

GUIDE: Microwave cannot be opened if the microwave is toggled on.

# Current Action Plan

        [
            "Microwave",
            "OpenObject"
        ],
        [
            "Mug",
            "PickupObject"
        ],
        [
            "Microwave",
            "CloseObject"
        ],
        [
            "SinkBasin",
            "PutObject"
        ],
        [
            "Faucet",
            "ToggleObjectOn"
        ],
        [
            "Faucet",
            "ToggleObjectOff"
        ]

# Feedback Information

After checking, the microwave's image is confirmed to be in the "
    ToggleOn" state.

### OUTPUT
```

```
        [
            "Microwave",
            "ToggleObjectOff"
        ],
        [
            "Microwave",
            "OpenObject"
        ],

        [
            "Mug",
            "PickupObject"
        ],
        [
            "Microwave",
            "CloseObject"
        ],
        [
            "SinkBasin",
            "PutObject"
        ],
        [
            "Faucet",
            "ToggleObjectOn"
        ],
        [
            "Faucet",
            "ToggleObjectOff"
        ]
```

Listing 3: **Example of the input (*i.e.*, Prompt) and the output (*i.e.*, Revised Action Plan) of the LLM.** This example describes the result where APM is applied. Compared to the current action plan, the action (*i.e.*, toggle off the microwave) is added in the revised action plan to avoid interaction failure.

```
### INPUT

# System Prompt
You need to determine whether to revise the action sequence to solve
    the task considering 'GUIDE' and then write down the final
    sequence of actions if needed.
This action sequence is made for solving household task. Each action's
     format is a triplet.
There are two options.
First, if you think that the first action is not needed, remove that
    action(triplet) and write down the actions after that.
You can only revise the first action; do not modify the actions (from
    the second actions) that follow, and write them down as originally
     provided.
In this context, 'action' refers to each element in triplet format in
    the list(action sequence).
Second, if you still think the first action is necessary, write it
    down exactly as received.
Do not add additional explanation. Just write the final action
    sequence in the right format (triplets in the list).

GUIDE: Our purpose is to perform the task efficiently.

# Current Action Plan

        [
            "Clean",
            "Target",
            "None"
        ],
        [
            "Move",
            "Bread",
            "Toaster"
        ],
        [
            "Move",
            "Bread",
            "Toaster"
        ],
        [
            "ToggleOn",
            "Toaster",
            "None"
        ],
        [
            "Move",
            "Bread",
            "Plate"
        ],
        [
            "Move",
            "Bread",
            "Plate"
        ]

# Feedback Information
The target object in the test image has been either cleaned, as
    determined by the similarity check.

### OUTPUT

        [
            "Move",
```

```
            "Bread",
            "Toaster"
        ],
        [
            "Move",
            "Bread",
            "Toaster"
        ],
        [
            "ToggleOn",
            "Toaster",
            "None"
        ],
        [
            "Move",
            "Bread",
            "Plate"
        ],
        [
            "Move",
            "Bread",
            "Plate"
        ]
```

Listing 4: **Example of the input (*i.e.*, Prompt) and the output (*i.e.*, Revised Action Plan) of LLM.** This example describes the result where APM is applied. Compared to the current action plan, the action (*i.e.*, clean the target) is deleted to skip an unnecessary action. A Target will be replaced with the corresponding object based on the context.

```
### INPUT

# System Prompt
You need to determine whether to revise the action sequence to solve
    the task considering 'GUIDE' and then write down the final
    sequence of actions if needed.
This action sequence is made for solving household task. Each action's
     format is a triplet.
There are two options.
First, if you think that the first action is not needed, remove that
    action(triplet) and write down the actions after that.
You can only revise the first action; do not modify the actions (from
    the second actions) that follow, and write them down as originally
     provided.
In this context, 'action' refers to each element in triplet format in
    the list(action sequence).
Second, if you still think the first action is necessary, write it
    down exactly as received.
Do not add additional explanation. Just write the final action
    sequence in the right format (triplets in the list).

GUIDE: There can be some objects that are already located in the
    desired destination.
If you think executing the following action should be avoided as it is
     no longer needed, add a Pass action in triplet form (same as
    given action) with Pass for action, None for Target and Parent,
    before the given action (including given action) without further
    explanation. If you think the following actions are still needed,
    repeat the given actions.

# Current Action Plan

        [
            "Move",
            "Target",
            "Parent"
        ]

# Feedback Information
After checking, it is found that the second object has already been
    placed in the desired location as indicated by the interaction
    mask, so no further interaction is needed.

### OUTPUT

        [
            "Pass",
            "None",
            "None"
        ],
        [
            "Move",
            "Target",
            "Parent"
        ]
```

Listing 5: **Example of the input (*i.e.*, Prompt) and the output (*i.e.*, Revised Action Plan) of LLM.** This example describes the result where ASR is applied. Given the current action plan (*i.e.*, move a target to a recep), LLM adds 'Pass' action in triplet which leads to skipping the ongoing interaction before the given action. Then the agent will not interact with the target since it is already located in the desired place as explained in the GUIDE. Target and Parent will be replaced with each corresponding object based on the context.

```
Driver tries to solve the task. Commander gives information helpful to
    solve the task. You have to get information through the dialog.
    Find the initial states of the objects and summarize them into a
    dictionary. If you cannot find proper information in the dialog,
    you should answer 'X'.

Just write a dictionary without giving an additional explanation. In a
    dictionary, fix the keys same with the example answers. Each of
    the keys has properties(keys) e.g., "location" which contains the
    initial location of the object. In some tasks, the driver may be
    asked to find multiple objects of one kind. In those cases, if an
    object is on Cabinet and another object is on CounterTop, you
    should output ['Cabinet', 'CounterTop']. (If you cannot know where
     the potato is initially, answer 'X'.), "receptacle" which is the
    place that the object should be placed on or in ultimately. "
    quantity" which represents the number of the objects, "
    quantity_of_slices" which represents the number of the object's
    slices. Some objects (e.g., plate, mug) have a key i.e., "Cleaned"
     which represents whether the object should be cleaned (then write
     "T") or not (then write "F"). If you cannot find the proper
    information, just write 'X'.

Dialog: {DIALOG}
Answer:
```

Listing 6: **Prompt for Extracting Task-Relevant Contexts from the Dialog.** {} denotes the section in the prompt that is replaced to each corresponding data. This prompt is designed to extract the useful information from the dialog to use it when making an initial action plan.

```
# System Prompt
You need to determine whether to revise the action sequence to solve
    the task considering 'INFO' and then write down the final sequence
     of actions if needed.
This action sequence is made for solving household task. Each action's
     format is tuple.
The possible action spaces are as follows.
['Target', 'PickupObject'], ['Target', 'PutObject'], ['Target', '
    OpenObject'], ['Target', 'CloseObject'], ['Target', '
    ToggleObjectOn'], ['Target', 'ToggleObjectOff'], ['Target', '
    PourObject'], ['Target', 'SliceObject']
For example, ['Target', 'PickupObject'] is a command to pick up the
    target.
Modify the action sequence by adding or subtracting action to suit the
     situation if needed. If it is not needed to be revised, just
    write down given action sequence.
In this context, 'action' refers to each element in tuple format in
    the list(action sequence).
You can add or subtract an action at the very first part of the action
     sequence if needed. Do not modify the actions that follow, and
    write them down as originally provided.
Do not add additional explannation. Just write the final action
    sequence in the right format (tuples in list).

{GUIDE}

# Current Action Plan
{CURRENTACTIONPLAN}

# Feedback Information
{FEEDBACK}
```

Listing 7: **Prompt for Revising Action Plan.** {} denotes the sections in the prompt that are replaced to each corresponding data. The prompt is designed to envelope the system prompt which gives the overall guideline for the LLM's task, the current action plan which is the source plan, and the feedback information which contains the feedback from the environmental discrepancy.

```
### INPUT

# System Prompt
You need to interpret conditional statements and write the
    corresponding situations as natural language prompts.
Based on the given code, describe in one sentence what the situation
    is when the conditional statement is true.
Do not start the sentence with 'The situation is ~'.

# Code
# test_img is a target object
# If Cleaned is true, it means that the target object is already
    cleaned.
Cleaned = self.check_similarity(test_img, "Clean", obj_name=
    planner_inputs['list_of_actions'][self.pointer-1][0])
    if Cleaned == 1:

### OUTPUT

The target object in the test image has been either cleaned, as
    determined by the similarity check.
```

Listing 8: **Example of Generated Feedback Prompt.** This example shows the generated feedback prompt by LLM. It is instructed to output the feedback prompt by providing a code used to detect discrepancy with description. Output is the feedback prompt.

