# OpenReview forum: "Pre-emptive Action Revision by Environmental Feedback for Embodied Instruction Following Agents"
_robot-learning.org/CoRL/2024/Conference — CoRL 2024_

### Official Review · Reviewer_Wspw · 2024-07-20
**Well-motivated paper, but the approach seems too specific. EDIT: The rebuttal addresses my concerns.**

**Originality:** 2
**Technical Quality:** 2
**Clarity Of Presentation:** 3
**Potential Impact:** 2
**Recommendation:** 3
**Confidence:** 3

**Review:**

**Strengths:**

* The paper addresses an important problem in embodied AI, namely, adapting an agent’s plan to remove discrepancy between the plan and the environment, and proposes a novel approach for it.
* Experimental evaluation shows significant improvement over 2 benchmarks — ALFRED and TEACh.


**Limitations:**

The main limitation of the approach seems to be that it is too specific in multiple ways:

* The authors propose 4 types of discrepancies, and a corresponding module to fix that problem. Thus, to make the system robust to errors that are not handled by the current approach would require defining more discrepancy classes and modules, which may not be scalable (particularly as these discrepancies and modules might interact with each other).
* Line 120 (main paper): In this example, the agent could find another potato before checking the refrigerator and complete the task faster, but isn’t this incorrect if the user wanted to use the one in the refrigerator? This may complete the task faster, improving results on the benchmarks, but does not solve the desired task.
* Lines 75-82 in the appendix describe the appearance detector module. This seems to be very simulator-specific — if the i_0 viewpoint is not a top-down view, would this step be needed?


**Other questions/issues:**


* What happens if the new modules proposed in this approach are not very good? E.g. the object attribute detector gives an incorrect prediction. Can this introduce errors into an already correct plan, instead of fixing them?
* The proposed approach should depend on how the initial plan was generated. Presumably, for any base approach that generates a plan, this approach can be applied on top to improve it. It would be good to see how the base approach used in this work performs without any of these modifications (the ablations in Table 3 addresses this to some extent, but it would be good to see the numbers without any of the additional modules). Further, if this approach was applied to other baselines, would that lead to further gains?



**Typos:**

Appendix Line 100: adjust → adjacent


EDIT:
In the rebuttal period, the authors applied the approach to 2 other baselines, and showed improvements on both. This addresses my concern that the approach might be too specific, and I've updated my rating from Weak Reject to Weak Accept.

**Quality Of The Limitations Section:**

1

**Questions For Rebuttal:**

- Line 126: I’m not sure I understand the following paragraph: does “o” map to potato, “Z_t” to table in the example?

- Line 170: How is the prompt e.g. “After checking, it appears that the target has already been cleaned” generated? Section B.3 in the appendix only describes how the attribute is predicted, but how is that information translated into language?

**Robotics Focus:**

3

**Summary Of Paper:**

This paper proposes an approach for embodied instruction following to dynamically update the plan based on environment feedback, using LLMs. They look at 4 types of discrepancies, and propose respective modules to fix each one.

**Summary Of Recommendation:**

The paper proposes a novel approach in embodied AI but the approach faces scalability concerns due to specificity in handling discrepancies and potential dependencies on simulator-specific modules, raising questions on robustness and broader applicability. EDIT: The authors applied the approach on other baselines during the rebuttal period, and have shown improvements, addressing my concerns about it being too specific.

---

### Official Review · Reviewer_nukf · 2024-07-22
**review for submission 58**

**Originality:** 2
**Technical Quality:** 2
**Clarity Of Presentation:** 4
**Potential Impact:** 2
**Recommendation:** 2
**Confidence:** 4

**Review:**

In this paper, the authors propose a RED framework which checks four distinct environmental discrepancies concerning objects to generate/revise actions for the agent to complete an embodied instruction following task. The logic of the paper is clear and easy to follow. Here are some concerns.

- Based on the paper, the environmental discrepancies are all predicted by some pretrained perception model, mostly relying on some vision models. However, there are attributes, appearance, etc, which might be difficult to recognize with only visual information. For example, how could the perception model recognize if an apple is cooked or not. Also, is there any mechanism to deal with the situation where the pretrained perception model fails.

- There are already a lot of works that try to revise or replanning action sequence based on large language models. What is the specific differences between the proposed method and other methods, such as (just name a few):
[1] ReAct: Synergizing Reasoning and Acting in Language Models
[2] ADaPT: As-Needed Decomposition and Planning with Language Models
The environmental feedback seems also included in other methods. I suggest a more thorough investigation of similar work and clarify the novelty of the proposed method. I am also curious about the performance when using a vision-language model for the task.

- It will be interesting to discuss the performance of the proposed method when it is applied in the real world. A real world demonstration will be a big plus to this work.

**Quality Of The Limitations Section:**

2

**Questions For Rebuttal:**

- Pretrained perception model limitation: Based on the paper, the environmental discrepancies are all predicted by some pretrained perception model, mostly relying on some vision models. However, there are attributes, appearance, etc, which might be difficult to recognize with only visual information. For example, how could the perception model recognize if an apple is cooked or not. Also, is there any mechanism to deal with the situation where the pretrained perception model fails.

- Difference and novelty of the proposed method with other methods: There are already a lot of works that try to revise or replanning action sequence based on large language models. What is the specific differences between the proposed method and other methods, such as (just name a few):
[1] ReAct: Synergizing Reasoning and Acting in Language Models
[2] ADaPT: As-Needed Decomposition and Planning with Language Models
The environmental feedback seems also included in other methods. I suggest a more thorough investigation of similar work and clarify the novelty of the proposed method. I am also curious about the performance when using a vision-language model for the task.

- Real-world evaluation: It will be interesting to discuss the performance of the proposed method when it is applied in the real world. A real world demonstration will be a big plus to this work.

**Robotics Focus:**

3

**Summary Of Paper:**

In this paper, the authors propose a RED framework which checks four distinct environmental discrepancies concerning objects to generate/revise actions for the agent to complete an embodied instruction following task.

**Summary Of Recommendation:**

There is no real-world evaluation and the novelty of the proposed method is a little weak.

---

### Official Review · Reviewer_M1LD · 2024-07-30
**Good paper to read, with a lot of experiments**

**Originality:** 3
**Technical Quality:** 4
**Clarity Of Presentation:** 4
**Potential Impact:** 3
**Recommendation:** 4
**Confidence:** 4

**Review:**

Strengths:
1. Clarity and Readability: The paper excels in its clarity and ease of understanding. The writing is straightforward and well-structured. The logical flow of the content ensures that key concepts are effectively communicated.
2. Innovative Methodology: The methodology presented in the paper is notably novel and introduces fresh approaches to the field. I did not see any paper in the field that talks about how to deal with such cases. Most of the papers focus on planning, not re-planning.
3. Comprehensive Experiments: The experiments conducted are thorough and cover a broad spectrum of scenarios. The detailed experimental setup and rigorous testing contribute to the robustness of the research findings.

Weaknesses:
1. Lack of Real-World Robotics Experiments: One notable limitation of the paper is the absence of real-world robotics experiments. While the simulation-based experiments provide valuable insights, the lack of practical application in real-world settings means that the findings may not fully address the challenges encountered in actual robotic environments. For instance, how to deal with the conditions besides the four listed in the paper.

**Quality Of The Limitations Section:**

3

**Questions For Rebuttal:**

1. One question I have is that how do you get the images of the plate before doing the task. I can imagine that you hardcode the image of the plate for each task, and assume that this is known. However, if the agent does not have such information before hand, how do you detect the the difference?

**Robotics Focus:**

3

**Summary Of Paper:**

The paper studies the problem in the instruction following, in which the agent is tasked with the instruction and is required to complete the task. The paper focuses on how the agent can react to unexpected conditions in the environment, and adaptively change the plan. In the paper, there are four unexpected conditions -- object presence, object appearance, object attributes, and object-object relationships. The paper proposes several methods to deal with these cases. The method is being verified in several embodied AI tasks (TEACh and ALFRED), and results indicate the performance is good.

**Summary Of Recommendation:**

The quality of the paper is good, and the easy to read. The method seems to be novel.

---

### Author Rebuttal · Authors · 2024-08-10

We thank the reviewers for their constructive feedback and encouraging remarks on excellent and well-structued presentation (**M1LD**, **nukf**), a novel and innovative approach (**M1LD**, **Wspw**), comprehensive and strong experiments (**M1LD**, **Wspw**) by tackling an important problem in embodied AI (**Wspw**). We address reviewers’ specific concerns as below. We note our revised contents in $\text{\color{red}red}$ in the revision.

---

### Decision · Program_Chairs · 2024-09-04

**Decision:**

Accept

**Comment:**

The submission introduces an approach for dynamic replanning for an agent's actions based on real-time environmental feedback. The proposed method reacts adaptively when discrepancies, such as object presence, appearance, attributes, and relationships, between the planned actions and the actual environment are detected. It is shown to be quite effective in 2 benchmarks (ALFRED and TEACh). On the other hand, the reviewers were concerned on the scalability and real-world applicability. The proposed method is only tested on simulated environments, which are even with relatively weak physical simulation, leaving questions about its effectiveness in complex scenarios and real world.

While it is indeed true that there is a need for further investigation, I recommend accept with the novelty of the proposed approach and its significant improvement. The learning based dynamic replanning, especially preventive one, is timely a good topic to discuss in the community. I however strongly suggest some real world demo in this case, which would enhance the paper quality and impact.